# The Antiproliferative Activity of a Mixture of Peptide and Oligosaccharide Extracts Obtained from Defatted Rapeseed Meal on Breast Cancer Cells and Human Fibroblasts

**DOI:** 10.3390/foods12020253

**Published:** 2023-01-05

**Authors:** Romina Lis Ferrero, Caroline Ruth Weinstein-Oppenheimer, Zaida Cabrera-Muñoz, María Elvira Zúñiga-Hansen

**Affiliations:** 1Escuela de Ingeniería Bioquímica, Pontificia Universidad Católica de Valparaíso, Av. Brasil 2085, Valparaíso 2362803, Chile; 2Escuela de Química y Farmacia, Facultad de Farmacia, Universidad de Valparaíso, Gran Bretaña 1093, Playa Ancha, Valparaíso 2360134, Chile; 3Centro de Investigación Farmacopea Chilena, Santa Marta 183, Playa Ancha, Valparaíso 2360134, Chile; 4Centro Regional de Estudio en Alimentos Saludables, R17A10001, Av. Universidad 330, Curauma, Valparaíso 2360134, Chile

**Keywords:** antiproliferative activity, mixture bioactive compounds, human fibroblasts, MCF-7, rapeseed

## Abstract

Oligosaccharide and peptide extracts obtained separately from defatted rapeseed meal (DRM) have shown antiproliferative activities on the MCF-7 breast cancer cell line. However, oligosaccharide extracts were not tested on human fibroblasts and have low yields. The objective of the present study was to combine two antiproliferative extracts, the peptides and oligosaccharides, that were obtained independently with commercial enzymes from DRM, allowing improvement of the mass yield and antiproliferative activity. The DRM was solubilized in an alkaline medium to obtain an insoluble meal residue (IMR) and an alkaline extract (RAE). To produce the oligosaccharide extract from IMR, three enzymes and different enzyme/substrate ratios were used. The oligosaccharide extract (molecular weight <30 kDa) recovered with the commercial enzyme. Endogalacturonase showed an 80% inhibition on MCF-7 cells at 20 mg/mL. The combination of this oligosaccharide extract with the peptide extract (obtained with Alkalase 2.4 L from a RAE at 10 mg/mL) inhibited 84.3% of MCF-7 cells proliferation at a concentration of 20 mg/mL, exhibiting no cytotoxic effects on fibroblasts. The mass yield of the extract pool was 27.07% (based on initial DRM). It can be concluded that a mixture of antiproliferative extracts was produced from DRM which was selective against MCF-7 cells.

## 1. Introduction 

Breast cancer today is a threat for women’s health, accounting for one in eight cancer diagnoses and a total of 2.3 million new cases. In 2020, it was, by far, the most commonly diagnosed cancer in women, and its burden has been growing worldwide, particularly in transitioning countries. New treatments are required due to a previously insufficient public health response to this development [1]. 

There is scientific evidence that suggests that protein hydrolysates and peptides independently derived from food may be important in reducing the cancer risk, which explains the significant interest that this area of research currently receives [2,3,4,5,6,7]. On the other hand, ingested pectins contribute to the soluble dietary fiber, and both hydrolyzed and non-hydrolyzed pectin extracts have shown cancer prevention and treatment properties [2,3,4,5,6,7]. 

A potential antitumor effect has been shown for rapeseed meal peptides on various cancer cells, including: cervical (HeLa): peptides recovered by enzymatic hydrolysis with Alcalase 2.4 L and Flavourzyme from an isolate of DRM albumin (protein obtained with a saline solution) [8]; liver cancer cells (HepG2): peptide (Trp-Thr-Pro (408.2 Da)) recovered by solid state fermentation, using the cooperation of mixed bacteria and a neutral protease [9,10]; and breast cancer cells (MCF-7)), previously recovered by the authors of this research [11]. In this investigation, an antiproliferative peptide extract was recovered from a DRM protein isolate obtained under alkaline conditions and the Alkalase 2.4 L protease, demonstrating an 83.9% inhibition of MCF-7 cells proliferation at 10 mg/mL, which is very high. Moreover, this effect was accompanied with no significant cytotoxicity on normal human fibroblasts.

DRM pectin shows a high content of galacturonic acid and arabinan and galactan sugars [12]. Galactan is composed of L-galactoses linked by β-(1,4) glycosidic bonds. Arabinan is composed of L-arabinoses linked by α-(1,5) bonds and are usually substituted by α-L-Arabinosa- residues (1,2)- and α-L-Arabinose-(1,3)- and/or α-L-Arabinose-(1,3)- and α-L-Arabinose-(1,3). Both galactans, like arabinans, are linked to ferulic acid via an ester bond [13]. The effects of pectin and/or its DRM hydrolysates on the inhibition of MCF-7 cells proliferation have been previously shown [14]. In the last report, pectin was extracted using four key treatments: alkalized water, EDTA, proteases (Alkalase 2.4L) and pectinases (Viscozyme L, Pectinex Ultra SP-L). The oligosaccharide extract recovered at the termination of the last stage of the process with commercial pectinases inhibited around 60% of cell proliferation, although with low mass yield from its original raw material (1.63% *w*/*w*) [14]. Therefore, low pectin yield and the lack of testing their cytotoxicity on healthy cells was an obstacle to progress on their further use as an anti-cancer agent. Accordingly, the present study raises the opportunity to design a recovery strategy for a peptide extract and, in parallel, an oligosaccharide extract, in this case through the use of commercial enzymes with different enzymatic activities, such as Endogalacturonase, Viscozyme L, Rohapect DA6L on Insoluble Meal Residue (IMR), with the aim of evaluating whether the combination of both extracts could, in addition to improving mass yield, maintain and/or improve their antiproliferative activity.

In addition, other authors (Maxwell et al., 2016; Li et al., 2019) [15,16] showed that oligopectins have demonstrated anticancer properties and not monomers. According to (Combo et al., 2012) [17], reaction times must be controlled to avoid the production of large amounts of monomers. These same authors suggested that the yields for oligopectin became lower at 2 h of reaction. Therefore, to obtain oligopectins enzymatically in the present study, an arduous selection of the enzyme preparation and its reaction conditions was required.

On the other hand, there are no precedents for the antiproliferative activity of mixtures of peptide and oligosaccharide extracts obtained enzymatically from DRM on MCF-7 and healthy cells.

The main aim of this paper was to obtain a mixture composed of peptide and oligosaccharide extracts of defatted rapeseed meal that inhibits MCF-7 cells proliferation but not healthy human cells to increase the mass yield and obtain the same or greater antiproliferative activity on MCF-7 by combining the two active fractions (peptide and oligosaccharide) recovered in parallel.

## 2. Materials and Methods 

### 2.1. Raw Materials 

The MCF-7 breast cancer cell line was obtained from ATCC, Virginia, USA (ATCC^®^ HTB-22™), and primary cultures of human fibroblasts were a kind donation from Inbiocriotec SA (Chile). Rapeseed cake (*Brassica napus*) was a gift from *Molinera Gorbea* (Temuco, Chile). Its handling was as shown in [11]. The cells were grown in DMEM (GE, Healthcare), supplemented with 10% Fetal Bovine Serum (Biological Industries, Beit HaEmek, Israel), penicillin/streptomycin, 100 UI/mL/10 µg/mL (ThermoScientific Inc^®^, Waltham, MA, USA) and 2 mM Glutamax (Thermo Scientific Inc^®^, Waltham, MA, USA). Alkalase 2.4 L FG was purchased from Novozymes (Bagsværd, Denmark), Viscozyme L (*Aspergillus aculeatus)* from Novozymes (Bagsværd, Denmark), Rohapect DA6L (*Aspergillus niger*) from AB-Enzymes (Darmstadt, Germany), and Endogalacturonase (*Aspergillus* sp.) was purchased from Biocatalysts (Wales, UK). 

The DRM characterization of dry matter, protein, crude fiber, fat and ash were performed before and after enzymatic treatment of DRM, according to the protocols described in AOAC (1990) [18] as shown in [11]. 

Carbonell’s assay was used for pectin content estimation [19]. Cell wall composition of DRM (cellulose, hemicellulose and lignin) was analyzed by the detergent fibre methods, developed by Van Soest [20] (Table 1). The reducing sugars in the hydrolysates after enzymatic treatment were determined using 3,5-dinitrosalicylic acid (DNS) method [21], and total carbohydrates by phenol-sulfuric acid assay [22], both using glucose as the standard material. 

In order to analyze the effect of the size of the molecules on the inhibition of cellular proliferation, 30 kDa membranes were used to fractionate the oligosaccharide extract. This molecular size was selected based on preliminary studies in the authors’ laboratory.

RPI and IMR from DRM were obtained as indicated in (Figure 1). RPI and the peptide extract were obtained by the same authors [11], and this part of the process is indicated with dotted lines. 

### 2.2. Extraction of Enzymatic Hydrolysates

Alkalase 2.4 L was used to hydrolyze RPI as previously described [11]. The mass yield of the peptide extract obtained with Alkalase 2.4 L from the protein isolate was 22.91% on the initial DRM. The Kjeldhal method was applied for protein determination.

Viscozyme L, Rohapect DA6L and an Endogalacturonase were used for IMR hydrolysis, which was carried out in batches in a 50 mL reactor (50 °C, pH 4). It was added to a measured amount of enzyme previously suspended in acetate buffer (100 mM) at pH 4 to obtain an enzyme/protein (E/S) ratio of 1, 3 and 5% (*w*/*w* (wet weight of the enzyme preparation/dry weight of solid on the substrate)). Solid/Liquid (S/L) ratios 1/10, 1/20 and 1/30 (ex: 1 g of IMR in 10 mL of acetate buffer at the corresponding pH) and hydrolysis times of 6, 9, 15, and 24 h were used. Also, control experiments were performed in which no enzyme was added. Heating of the reaction mixture at 90 °C for 5 min, followed by a 5 min ice bath incubation was used to stop hydrolysis. Once the S/L and E/S ratio had been defined, through determination of soluble solids and concentration of total and reducing sugars at each hydrolysis time, the effect of three temperatures (40, 45, 50 °C) and three pH (4, 4.5 and 5) on the hydrolysis of IMR with the different commercial enzyme preparations was examined. The hydrolysates were then frozen at −80 °C and later freeze-dried. 

### 2.3. Antiproliferative Activity

The antiproliferative activity of MCF-7 or normal human cells of the extract peptide, oligosaccharide and/or the combination of the extracts were examined using the resazurin viability assay [23]. For that assay, dried extracts were suspended in supplemented DMEM culture media, and the viability of the cells was determined using concentrations of 3.75 to 20 mg/mL. In addition, the IC50 concentrations of all the extracts were calculated in an Excel table.

The combination of the extracts was made by the mixing the total of each recovered extract, from 100 g of DRM according to the mass balance presented in Figure 1. As observed, 21.54 g total solids were recovered in the peptide extract (E1) and 3.91 g in the oligosaccharide extract (E2). Based on the total mass recovered, the mixture was carried out, in order to use 100% of each recovered extract. Therefore, it was made up of 84.6% E1 and 15.4% E2. The mixture was dissolved in the DMEM medium and left under gentle stirring for 12 h for its complete dissolution and measurement of cell proliferation inhibition. 

### 2.4. Ultrafiltration

The filtration of oligosaccharide extract was evaluated in an Amicon-stirred ultrafiltration cell (Millipore Corporation, Darmstadt, Germany) (MWCO of 30 kDa) at standard temperature and pressure (25 °C, 1 atm). Permeate and retained samples were collected, lyophilized, and stored in a desiccator at room temperature (25 °C). 

Initial Oligosaccharide hydrolyzed extract were characterized by the DNS method and the total carbohydrate method (both using glucose as the calibration standard). In addition, the protein content was evaluated by the Kjeldhal method and minerals as its ash content (Table 2). After filtration, the fraction’s antiproliferative activity was determined, and the recovery yield was reported as % mg solid in the corresponding fraction/ mg total solid in the original extract (Table 3).

### 2.5. Statistical Analysis

All analytical determinations were performed in triplicate. Values were expressed as the mean ± standard deviation (*n* = 3). Analysis of variance was conducted, and differences between variables were tested for significance by one-way analysis of variance using Minitab (Version 21, Minitab Inc, State College, PA, USA). A significant difference was determined with a 95% confidence interval (*p* < 0.05). 

## 3. Results

### 3.1. Characterization of the Recovered Extracts

Figure 1 shows the approximate composition (g/100 g wet matter) of DRM, and Table 1 presents the composition of detergent fibers and pectin in DRM and IMR. DRM presented a high protein content and nitrogen-free extract, which is beneficial to recover peptides and pectin from this material. The values found are similar to those reported by other authors [24,25,26].

IMR is composed of 13.85% protein, 10.1% ash, 15.3% crude fiber and 50.7% ENN. The pectin analysis showed that 49.2% of the total pectin in the DRM remained in the IMR, while the rest was solubilized with the alkaline treatment of the DRM to recover the peptide extract, and after obtaining RPI, most of the pectin could remain soluble in SA, due to the pH of the medium (4.5). Although this pectin is solubilized in SA, it was possible to recover it by ethanol precipitation, however, its yields were low (around 1.5%), and it was not considered in the final antiproliferative extract. 

### 3.2. Selection of the Commercial Enzyme Preparations from IMR

The inhibition of MCF-7 cells proliferation of oligosaccharide extracts obtained with Endogalacturonase, Rohapect DA6L and Viscozyme L extracts from the IMR at 9 h is presented in Figure 2. The reaction time was selected based on previous studies at different reaction times that are not shown in this report. It was observed that the controls without enzyme did not inhibit the MCF-7 cells proliferation under the studied concentrations, indicating that the pH (4) and temperature (50 °C) of the reaction medium during the same hours of reaction did not influence the antiproliferative activity of the extract, being stable under such conditions and requiring enzymatic hydrolysis to recover bioactive compounds. The oligosaccharide extracts recovered with Endogalacturonase and Rohapect DA6L inhibited the proliferation of MCF-7 cells by 80% at a concentration of 20 mg/mL, although at lower doses (3.75 mg/mL), the Endogalacturonase extract was more effective among the two enzyme preparations, which could be related to the enzymatic activity of the enzymes and their secondary activities, pectinase and xylogalacturonase, respectively. In the case of Viscozyme L, a lower cellular inhibition was observed under the concentrations studied in comparison to the other two enzymes. Similar results were demonstrated by [23] who recovered a pectic extract of DRM with Viscozyme L (15 h reaction), with an antiproliferative activity on MCF-7 of around 60% at 10 mg/mL and without showing a dependence between the concentration of the extract and cell proliferation inhibition. In our research, a dependence between the concentration of oligosaccharide extract and cell inhibition was evidenced, which could be due to the different extraction process and different raw material supplier, which could generate extracts with different compound profiles. The named differences regarding the extraction processes are based on the authors of [23] who used a process based on the recovery of pectins, starting with an aqueous solubilization of DRM, then with EDTA, and finally an enzymatic hydrolysis, first with an Alkalase 2.4 L and then with Vizcozyme L. In the present study, the process was designed to recover proteins through an alkaline solubilization of DRM, and then the insoluble fraction of this treatment was used for the recovery of an oligosaccharide extract.

From a mass balance performed according to the extraction of an oligosaccharide hydrolysate (Table 2), the yield of soluble solids was 32.18%, 37.28% and 29.81% for Endogalacturonase, Viscozyme L and Rohapect DA6L, respectively, with respect to the total amount of solids in the IMR substrate. Therefore, Viscozyme L treatment extracted 7.5% more than Rohapect DA6L and 5.1% more than Endogalacturonase. These results are similar to those obtained by Rodrigues et al. 2014, who demonstrated that Viscozyme L (S/L 1/10 *w*/*v*, 45 °C and pH 3.5) was the most efficient biocatalyst for elimination of carbohydrates directly from rapeseed meal, with an extraction yield of reducing sugars of 80%, as equivalent units of glucose, related with its endo and exa-carbohydrase activity.

The total carbohydrate content revealed that Viscozyme L extracts had twice as many total sugars as the ones obtained with Endogalacturonase treatment, while Rohapect DA6L extracts had 1.3 times more than the ones recovered with Endogalacturonase. The opposite situation was observed with the content of reducing sugars, where Endogalacturonase extracts presented 1.3 times more than the ones obtained with Rohapect DA6L, although Viscozyme L was the enzyme that generated more reducing sugars, possibly due to a higher oxo activity of the enzyme. Considering that the monomers have not demonstrated antiproliferative activity, a low content of reducing sugars and a high content of total sugars are desirable. As it was expected that from the use of different enzymes with different activities, it was possible to obtain extracts with different characteristics. Endogalacturonase is a pectinase that randomly cleaves α-1,4-D-galacturonic acid, generating oligosaccharides [27]. According to the supplier, Rohapect DA6L is a polygalacturonase with strong arabanase activity. Viscozyme L is a multienzyme complex whose main activity is that of endoglucanase and as secondary cellulase (endo and exo), hemicellulase, xylanase and pectinase.

Therefore, considering the cellular inhibition on MCF-7 cells and the characterization of the extracts, the inhibition of healthy cells of the three oligosaccharide extracts obtained with Endogalacturonase, Viscozyme L and Rohapect DA6L at 9 h of hydrolysis from the IMR was evaluated for the selection of the enzyme preparation as shown in Figure 3.

As shown in Figure 3, all three extracts demonstrated a slight cytotoxicity on healthy human fibroblasts below 30%. Moreover, the extract obtained with Endogalacturonase did not show inhibition at low concentrations, while high doses showed the least effect, compared to the Viscozyme L and Rohapect DA6L extracts that showed some inhibition of human fibroblasts proliferation at all the examined concentrations. A recent study [28] showed that the Galactose and Arabinose content of the side chains in the Ramnogalacturonan-I/Homogalacturonan backbone of pectin have an antitumor effect on U251-MG cells, while it was not observed cytotoxicity in fibroblast cells (NIH-3T3). Therefore, a higher content of these neutral sugars in the extracts may not be cytotoxic to human fibroblasts.

The Endogalacturonase extract was selected, considering the results presented on the antiproliferative activity on MCF-7 and human fibroblasts of oligosaccharide extracts obtained from IMR, because it was the one that exhibited the higher inhibition of cancer cells and affected the least the healthy cells.

### 3.3. Optimization of Reaction Conditions with the Selected Enzyme Endogalacturonase

Previously, for the selection of the enzyme preparation, standardized reaction conditions were used, and thus Figure 4 shows the selection of the reaction conditions with Endogalacturanase from IMR. As shown in Figure 4a, the enzyme hydrolyzed the substrate efficiently compared to the control without enzyme. The initial reaction rate and the solubilization profiles of the IMR were similar for both S/L ratios of 1/20 and 1/30. A 1/20 S/L ratio was selected for the hydrolysis of the IMR because of the ease of handling the sample, for example during filtering, which is useful for large-scale applications and advisable to increase accessibility enzyme to substrate and the amount of IMR treated per volume of hydrolysis solution. The ratio S/L 1/10 was not homogeneous to take the samples, and for this reason it was not used in practice.

In Figure 4b it is shown that with an enzyme concentration of 1 and 3% *w*/*w*, almost 15% of the IMR is solubilized at 9 h of hydrolysis, while with a concentration of 5% *w*/*w* of enzyme a 16.5% IMR. Thus, this small difference does not justify spending five times more enzyme; therefore, an E/S ratio 1% *w*/*w* was selected.

The hydrolysis yield was 32.18% with respect to the solids in the IMR substrate at an E/S ratio 1% *w*/*w*, an S/L ratio 1/20 *w*/*v*, 50 °C and pH 4, while approximately 1.8% of the solids were solubilized in the control only due to the condition of 50 °C and pH 4, for 9 h.

### 3.4. Effect of the Mean Size of the Oligosaccaride Present in the Extract Obtained with Endogalacturonase from IMR on the Cellular Proliferation Inhibition of MCF-7 and Human Fibroblasts

The results presented in the Table 3 show the solid recovery yields in the separation with 30 kDa membranes of the oligosaccharide extract obtained with Endogalacturonase from the IMR.

The results obtained (Table 3) showed that 28.8% of the solids in the Initial Oligosaccharide Extract were eliminated in the filtrate prior to ultrafiltration (0.45 µm and 0.22 µm membranes). These high molecular weight compounds could be fibers, which, as seen previously in the characterization of the extract (Table 2), contained cellulose, hemicellulose and unhydrolyzed lignin. In addition, the previous filtration eliminated 3.9% of the proteins present in the Initial Oligosaccharide Extract. Regarding ultrafiltration, 47.5% of the solids were obtained in the Fraction I, while 23.7% showed a molecular weight of less than 30 kDa. This could indicate that the oligosaccharide extract recovered with Endogalacturonase has a greater part of oligosaccharide, hemicellulosic and cellulosic oligosaccharides with high molecular weight. As observed in Figure 4, a longer hydrolysis time does not improve the recovery of low molecular weight oligosaccharides (<30 kDa), confirming preliminary studies that 9 h of hydrolysis added to an ultrafiltration with 30 kDa membranes obtains an extract of oligosaccharides with bioactivity on MCF-7 breast cancer cells

The analysis of the antiproliferative activity on MCF-7 cells and human fibroblast of the different fractions obtained by ultrafiltration of the oligosaccharide extract with Endogalacturonase from IMR is shown in Figure 5. As can be seen (Figure 6a), the compounds with a molecular mass of less than 30 kDa (Endogalacturonase < 30 kDa) demonstrated a strong inhibitory effect on MCF-7, similar to the total extract without ultrafiltration (80% at 20 mg/mL). While those greater than 30 kDa (Endogalacturonase > 30 kDa) did not inhibit cancer cells under any of the concentrations studied. These results suggest that oligosaccharides less than 30 kDa have a high inhibitory effect on MCF-7 cells with a recovery yield of 23.71% (based on the initial oligosaccharide extract). These results coincide with other studies (Naquash et al., 2017; Ramos do Prado et al., 2019) [29,30] that reported that pectin with lower molecular mass produces greater antitumor activity than naturally large pectin. According to Ramos do Prado et al., 2019 [30], who fractionated citric pectin with membranes of 3, 10 and 30 kDa and analyzed their inhibition on colon (HCT116 and HT29) and prostate (PC3) cancer cells proliferation, suggested that the enrichment of de-esterified homogalacturonans oligomers and the depletion of type I arabinogalactans and rhamno-galacturonans in the fraction less than 3 kDa, or the increase of type I arabinogalactans and the loss of rhamno-galacturonans in the fraction between 10–30 kDa, improve anticancer behavior by inhibiting cancer cell migration, aggregation, and proliferation. On the other hand, it could be due to Napin peptides, since according to other authors (Ng et al., 2011) [31], a polypeptide (13.8 kDa) similar to Napin from *Brassica campestris* and *Brassica parachinensis* showed antiproliferative effects on leukemia cells (L1210), although there are no studies with Napin from *Brassica napus* that demonstrate inhibitory activity on MCF-7.

Figure 5b showed that the fraction called Endogalacturonase < 30 kDa, which did not show cytotoxicity on healthy human fibroblast cells, and Endogalacturonase > 30 kDa was slightly cytotoxic at high concentrations, the same as the Initial Oligosaccharide extract called Endogalacturonase in the graph. Thus, fractionation improved the selectivity of the extract containing low molecular weight olygopectin. It is appreciated that in such cases the error is higher; however, it is always maintained with a level of negative inhibition showing the low toxicity in such samples. According to (Amaral et al., 2019) [28], the content of Galactose and Arabinose of the side chains in the RG-I/HG backbone of pectins have an antitumor effect on U251-MG cells, while cytotoxicity was not observed in normal fibroblast cells (NIH-3T3). Therefore, a higher content of these neutral sugars in the extracts seems to be non-cytotoxic for human fibroblasts, as shown here.

### 3.5. Effect of the Combination of Oligosaccharide Extract and Peptide Extract Obtained from DRM on the Cellular Inhibition of MCF-7 and Human Fibroblasts Cells

Considering the previous results and those published in the authors’ previous research (Ferrero et al., 2021) [11], and with the objective of demonstrating synergistic, additive, or no effects in terms of antiproliferative activity and increase, the final mass yield of the extract in comparison to each of the individual extracts (peptide extract: 22.91%, oligosaccharide extract: 4.16%), maintaining cellular inhibition of at least 80% on MCF-7 cells without a cytotoxic effect on healthy human fibroblast cells, the effect of the combination of the peptide extract and the oligosaccharide extract on the cellular inhibition of MCF- 7 and human fibroblast was evaluated (Figure 6).

As shown in Figure 6a, the inhibition of MCF-7 cell proliferation from the combination of the antiproliferative extracts was 84.3% at 20 mg/mL and 82.4% at 10 mg/mL. These values were similar to those of the peptide extract, which could be expected since the mixture consisted mainly of this extract (85%). However, at 7.5 mg/mL, the antiproliferative activity of the combination of extract is lower than the peptide extract (60% and 75.4%, respectively), while at 5 mg/mL, the combination of extracts (oligosaccharide extract + peptide extract) showed a greater antiproliferative effect than the individual extracts. Therefore, it was demonstrated that the combination of antiproliferative extracts retained a potential anticancer effect on the MCF-7 cell line, greater than 80% at 10 and 20 mg/mL, and without a cytotoxic effect on healthy human fibroblast cells (Figure 6b).

The mass yield of the peptide extract obtained with Alkalase 2.4 L from the protein isolate was 22.91% on the initial DRM, and the mass yield of the oligosaccharide extract obtained from the meal residue with Endogalacturonase and ultrafiltration (fraction less than 30 kDa), was 4.16% with respect to the initial DRM. Therefore, the mass yield of the combination of extracts was 27.07% (with respect to 100 g of initial DRM), and no synergistic or additive effects were demonstrated between the peptide and oligosaccharide extracts in terms of cellular proliferation inhibition on MCF-7 cells. This could be due to the fact that the mixture was more concentrated in protein material since the amount of protein evaluated in initial DRM was 30.6%, compared to 6.56% of pectin. However, for industrial purposes, a detailed analysis of the costs of said process should be made and based on this variable and that it is a product with high added value, consider the profitability in the recovery of pectin.

In summary, Table 4 shows a comparison of IC50 (mg/mL) for the proliferation inhibition of MCF-7 and fibroblasts by the tested extracts, demonstrating that it was possible to obtain a mixture of the peptide and oligosaccharide extracts and that it showed the greatest antiproliferative effect on MCF-7 breast cancer cells, with no effect on human fibroblast cells.

## 4. Conclusions

The alkaline DRM treatment allowed the recovery of an IMR with a mass yield of 51.2% (based on the initial DRM). The enzymatic hydrolysis of IMR with commercial enzyme preparations such as Endogalacturonase, Rohapect DA6L and Viscozyme L allowed the recovery of an oligosaccharide extract with high cellular inhibition on MCF-7 cells, of 80% for the cases of Endogalacturonase and Rohapect DA6L at 20 mg/mL. However, a slight cytotoxic effect of these extracts was demonstrated on healthy human fibroblast cells, the oligosaccharide extract obtained with Endogalacturonase being the least cytotoxic. It was also shown that the oligosaccharide fraction less than 30 kDa of this last extract showed selectivity towards cancer cells, with 80% inhibition at a concentration of 20 mg/mL and a mass yield of 4.16% based on the initial DRM.

The results of this study indicate that it was possible to obtain an antiproliferative extract from DRM through the enzymatic recovery of a peptide and oligosaccharide extract and their combination, with a high cellular inhibition on MCF-7, high yield and without cytotoxic effects on human fibroblast cells. Therefore, their combination did not show additive and/or synergistic effects but improved the final mass yield of the process to formulate nutraceutical foods with antiproliferative properties in MCF-7 breast cancer cells.

## Figures and Tables

**Figure 1 foods-12-00253-f001:**
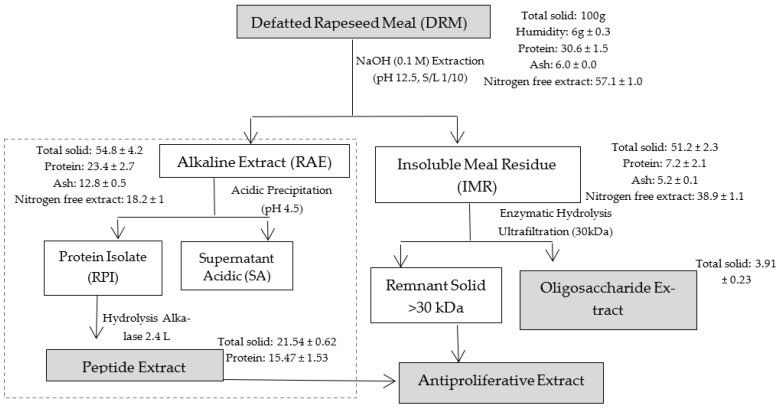
Workflow of antiproliferative extract extraction.

**Figure 2 foods-12-00253-f002:**
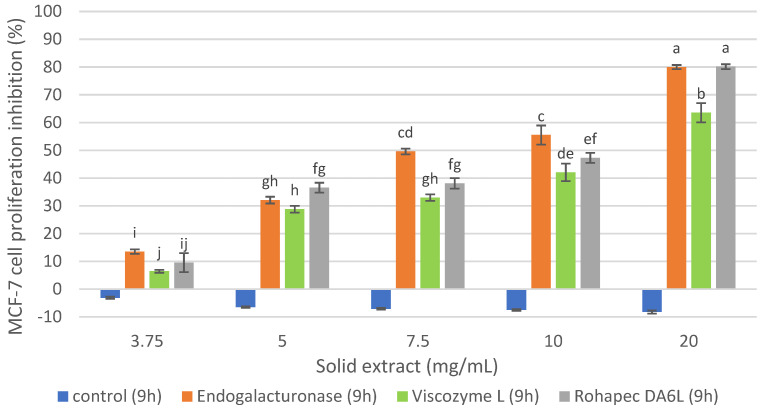
MCF-7 cell proliferation inhibition of the oligosaccharide extract obtained with commercial enzyme preparations from IRM at pH 4, 50 °C, S/L 1/20, E/S 1% p/p. Percentages of inhibition are calculated in reference to the signal of cells grown of extract-free cell culture media. Mean ± SD (*n* = 3). The median that does not share a letter are significantly different (*p* > 0.05), comparing all values together with ANOVA and Tukey’s test.

**Figure 3 foods-12-00253-f003:**
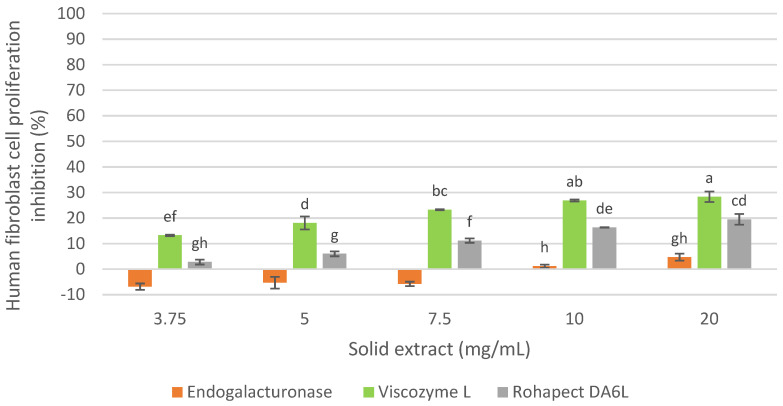
Human fibroblast cell proliferation inhibition of oligosaccharide extract obtained with commercial enzyme preparations from RM. Percentages of inhibition are calculated in reference to the signal of cells grown of extract-free cell culture media. Mean ± SD (*n* = 3). The median that does not share a letter are significantly different (*p* > 0.05), comparing all values together with ANOVA and Tukey’s test.

**Figure 4 foods-12-00253-f004:**
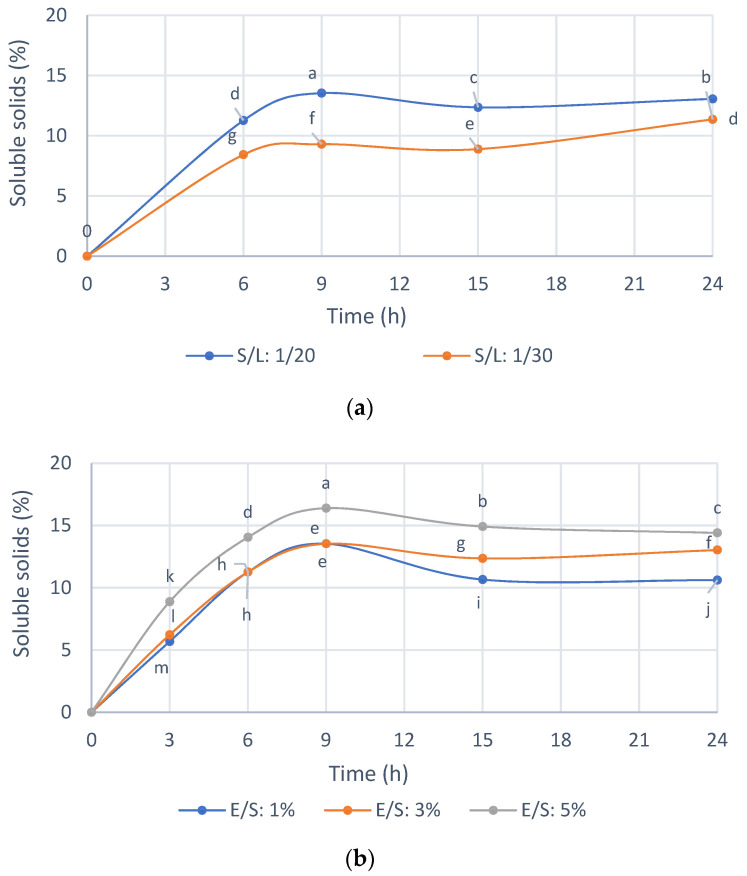
(**a**) Effect of Solid/Liquid ratio on the soluble solids with Endogalacturonase (50 °C, pH 4, E/S 3%). (**b**) Effect of enzyme/substrate ratio on the soluble solids with Endogalacturonase (50 °C, pH 4, S/L 1/20). E/S = wet weight of enzyme preparation/dry weight of solid of the substrate. Mean ± SD (*n* = 4). The median that does not share a letter are significantly different (*p* > 0.05), comparing all values together with ANOVA and Tukey’s test.

**Figure 5 foods-12-00253-f005:**
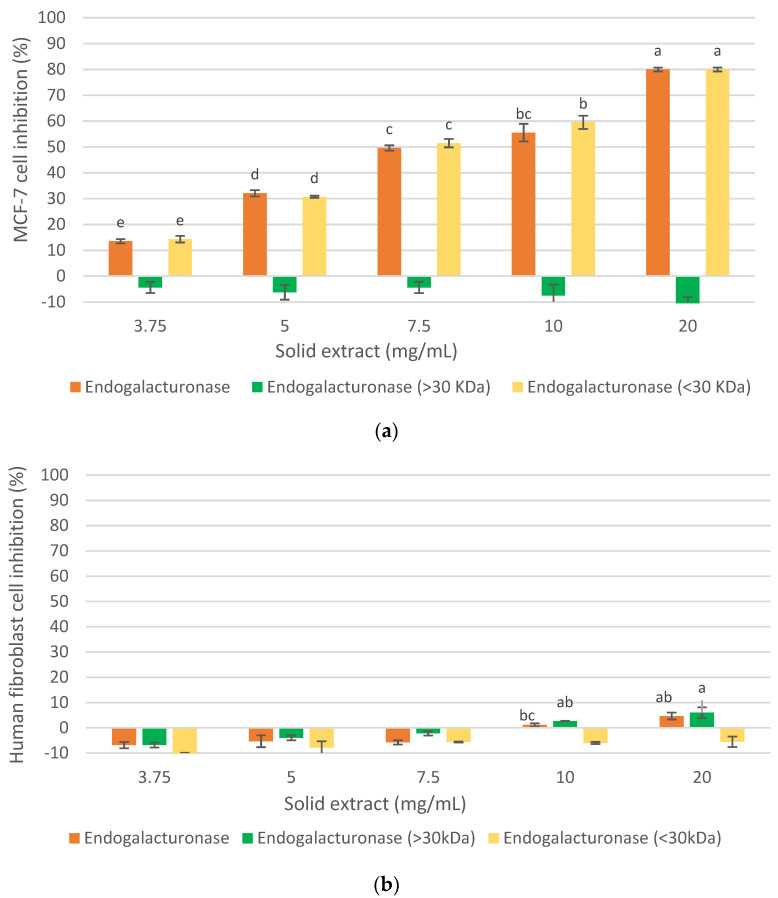
Effect of molecular weight of the fractions obtained from the oligosaccharide extract by ultrafiltration, on the cellular inhibition. (**a**) on MCF-7 cancer cell line, (**b**) on healthy human fibroblast cell. Mean ± SD (*n* = 3). The median that does not share a letter are significantly different (*p* > 0.05), comparing all values together with ANOVA and Tukey’s test.

**Figure 6 foods-12-00253-f006:**
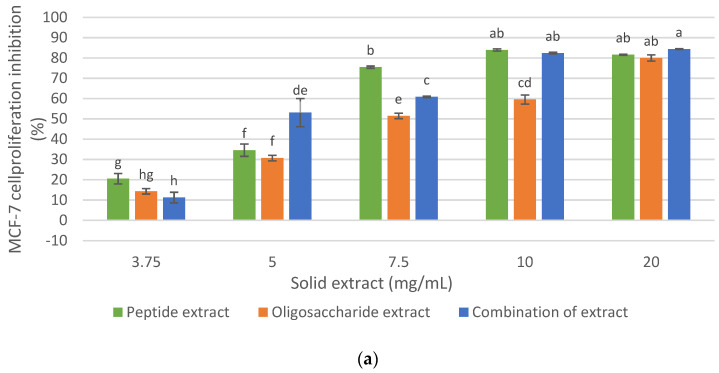
Effect of the combination of oligosaccharide extract and protein extract obtained from DRM on the cellular inhibition. (**a**) on MCF-7 cancer cell line. (**b**) on human fibroblast cell. Mean ± SD (*n* = 3). The median that does not share a letter are significantly different (*p* > 0.05), comparing all values together with ANOVA and Tukey’s test.

**Table 1 foods-12-00253-t001:** Composition of detergent fibers and Pectin in DRM and IMR.

	DRM	IMR
Cellulose (g)	9 ± 2	8.9 ± 0.9
Hemicellulose (g)	6.5 ± 0.5	5.0 ± 0.9
Lignin (g)	9.4 ± 0.4	7.0 ± 0.2
Pectin (g)	6.6 ± 0.5	3.0 ± 0.5

Mean ± SD (*n* = 3).

**Table 2 foods-12-00253-t002:** Characterization of oligosaccharide extract obtained with commercial enzyme preparations from IMR at pH 4, 50 °C, S/L 1/20, E/S 1% p/p.

	Hydrolysate-Endogalacturonase	Hydrolysate-Rohapect DA6L	Hydrolysate-Viscozyme L
Total solid (g)	16.5 ± 0.5	15.3 ± 0.2	19.1 ± 0.3
Protein (g)	3.4 ± 0.4	2.84 ± 0.4	5.1 ± 0.3
Ash (g)	3.2 ± 0.1	3.1 ± 0.2	3.9 ± 0.2
Reducing sugars (g/L)	5.26 ± 0.05	4.0 ± 0.5	10.92 ± 0.03
Total carbohydrates (g/L)	8.30 ± 0.04	10.78 ± 0.05	16.91 ± 0.04
	**Remnant Solid-Endogalacturonase**	**Remnant Solid-Rohapect DA6L**	**Remnant Solid-Viscozyme L**
Total solid (g)	34.7 ± 0.5	35.9 ± 0.6	32.1 ± 0.4
Pectin (g)	2.6 ± 0.2	2.4 ± 0.3	1.8 ± 0.4
Hemicellulose (g)	4.1 ± 0.6	4.5 ± 0.5	3.3 ± 0.6
Cellulose (g)	3.1 ± 0.7	1.1 ± 0.2	5.9 ± 0.5
Lignin (g)	1.6 ± 0.3	1.2 ± 0.3	1.8 ± 0.9

**Table 3 foods-12-00253-t003:** Oligosaccharide extract fractionation by ultrafiltration.

Fraction	Solids (g)	% Recovery of Solids (*w*/*w*) ^a^
Initial Oligosaccharide Extract	16.5 ± 0.3	
Filtered (0.45 and 0.22 µm)	4.8 ± 0.2	28.8
Fraction I (>30 kDa)	7.8 ± 0.2	47.5
Fraction II (<30 kDa)	3.9 ± 0.3	23.7

a. % mg solid in the corresponding fraction/ mg total solid in the Original Extract. Mean ± SD (*n* = 2).

**Table 4 foods-12-00253-t004:** Comparison of IC50 (mg/mL) for the proliferation inhibition of MCF-7 and fibroblasts by the tested extracts.

Treatment	MCF-7	Fibroblast	Sensitivity IC50_F_/IC50_MCF-7_)
Endogalacturonase	8.6	No inhibition observed	
Viscozyme L	12.2	169.0	14
Rohapect DA6L	9.5	330.3	35
Endogalacturonase > 30 kDa	No inhibition observed	No inhibition observed	
Endogalacturonase < 30 kDa	8.4	No inhibition observed	
Mixture	6.3	No inhibition observed	

## Data Availability

The data are available from the corresponding author.

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
