# Peer review of "The Antiproliferative Activity of a Mixture of Peptide and Oligosaccharide Extracts Obtained from Defatted Rapeseed Meal on Breast Cancer Cells and Human Fibroblasts"

_foods, 2023, doi:10.3390/foods12020253_

Round 1
Reviewer 1 Report
In results,
Figures 2 -5. Where is the figure legend, authors should put legends for different column colors? I suggest calculating the IC50 value
Figure 5, it seems that the value of standard error is high. please double check
Did the author investigate the components in the tested sample with promising activity with either LC/MS or GC/MS analyses, if this, what about performing a molecular docking study as tool for highlighting the virtual machines of activity
Did the author investigate or predict the effective molecular target at least from the reported literature?
Authors should highlight the logic behind choosing the MCF-7 (breast cancer)
Authors should perform appropriate statistical analyses in figures 2-5.
Author Response
I attach answers in word.

Reviewer 2 Report
Suggestions to Author/s
Dear Dr. Romina L. Ferrero, as a selected reviewer, I made the prompt check of your scientifically important article, and found it Good.
Strengths
1. The objective of the present study is to obtain a mixture consisting of peptide and oligosaccharide extracts of defatted rapeseed meal that inhibits a breast cancer cell line but not healthy human fibroblasts, which allows to increase the mass yield, and to obtain the same or greater antiproliferative activity on MCF-7 by combining the two active fractions (peptide and oligosaccharide) recovered in parallel.
2. The oligosaccharide and peptide extracts obtained separately from defatted rapeseed meal (DRM) demonstrated the antiproliferative activities on MCF-7 breast cancer cell line. However, oligosaccharide extracts were not tested on human fibroblasts and have low yields. The objective of the present study was to combine two antiproliferative extracts, the peptide and oligosaccharide, that were obtained separately with commercial enzymes from DRM, that allows to improve the mass yield and anticancer activity. The DRM was solubilized inanalkaline medium to obtain aninsoluble meal residue (IMR) and analkaline extract (RAE). To produce the oligosaccharide extract from IMR, three commercial enzyme preparations and different enzyme/substrate ratios were used. The oligosaccharide extract (molecular weight lower than 30 kDa) obtained with endogalacturonase showed an 80% inhibition on MCF-7 cells. The combination of this oligosaccharide extract with the peptide extract (obtained with Alkalase 2.4 L from a RAE) inhibited 84.3% of MCF7 cells in a concentration of 20 mg/mL, but without any cytotoxic effects on fibroblasts. The mass yield of the extract pool was 27.07% (based on initial DRM).
3. It can be concluded that it was possible to obtain a mixtureof antiproliferative extracts from DRM and being selective against MCF-7 cells, so providing a favorable knowledge in the search for new drugs against breast cancer. The alkaline DRM treatment allowed to recover an IMR with a mass yield of 51.2% (based on the initial DRM). The enzymatic hydrolysis of IMR with commercial enzyme preparations such as Endoga- lacturonase, Rohapect DA6L and Viscozyme L allowed to recover a oligosaccharide extract with high cellular inhibition on MCF-7 cells, of 80% for the cases of Endogalacturonase and Rohapect DA6L at 20 mg/mL. However, a slight cytotoxic effect of these extracts was demonstrated on healthy human fibroblast cells, the oligosaccharide extract obtained with Endogalacturonase being the least cytotoxic. It was also shown that the oligosaccharide fraction less than 30 kDa of this last extract showed selectivity towards cancer cells, with 80% inhibition at a concentration of 20 mg/mL and a mass yield of 4.16% based on the initial DRM. The results of this study indicate that it was possible to obtain an antiproliferative extract from DRM through the enzymatic recovery of a peptide and oligosaccharide extract and their combination, with a high cellular inhibition on MCF-7, high yield and without a cytotoxic effects on human fibroblast cells and therefore, their combination did not show additive and/or synergistic effects but improves the final mass yield of the process to formulate nutraceutical foods with antiproliferative properties in MCF-7 breast cancer cells.
Weaknesses
1. During the prompt check the English language problems were found. They are highlited with the red colour. Please correct them. So please make the additional English editing with the help of proper person.
2. During the prompt check of your article the figures were found to be improper. So, please redraw them in the different colours with the celar legend designation with different colours and/or symbols.
3. Please insert into the figures the statistical designations with the asterixs (1 2 or 3)
To do this, please contact the editor to avoid any problems with the additional redrownings.
4. Finnaly, a kind suggestion: due to the experimental quality and versality, you should prepare in addition an Europatent, which will give a broader dissemination of your results.

Author Response
I attach answers in word.

Reviewer 3 Report
Totally speaking, this research article regarding the oligosaccharide and peptide extracts obtained separately from defatted rapeseed meal (DRM), which have been demonstrated to have antiproliferative activities on the MCF-7 breast cancer cell line, is very well designed. The current study's goal was to combine two antiproliferative extracts – peptide and oligosaccharide – that were previously generated independently using DRM commercial enzymes in order to increase mass yield and antiproliferative efficacy. The DRM was solubilized in an alkaline medium to obtain: 1) insoluble meal residue and 2) alkaline extract. An insoluble meal residue (IMR) was used to produce oligosaccharide fractions/extracts, while the alkaline extract was utilized to produce peptide fractions/extracts. The combination of oligosaccharide extract with the peptide extract inhibited of MCF7 cells, without cytotoxic effects on fibroblasts. This study was suitable for Foods journal, and proposed special issue. However, there are a few points that require clarification.
(1) Lines 6 to 12: Please include the matching e-mail addresses of the co-authors listed above, as well as their initials in parenthesis.
(2) Abstract: The section describing the outcomes should be specified. Please provide the concentration of oligosaccharide extract if the concentration of 20 mg/mL peptide extract is given. In the abstract, be sure to specify which of the three enzyme preparations for the production of oligosaccharides is being discussed if you claim that there have been three. The same holds true for the creation of extracts abundant in peptides. Additionally, describe in detail which oligosaccharide extract and peptide fraction combination (which mass ratio of one to the other fraction) exhibits the most antiproliferative effects.
(3) Keywords: Because this study is about in vivo research at the level of cell lines, whether cancerous or non-cancerous, I believe it is inappropriate to use the term "anticancer," because these studies only look at antiproliferative activity. For the exact confirmation of anticancer activity, in those samples that exhibit high antiproliferative activity, it is necessary to conduct research on animals with cancerous cells.
(4) Introduction: Please complete the introduction with specific data on the oligosaccharide and/or peptide fractions from defatted rapeseed meal and their bioactive properties. Next, isolate the procedures for obtaining these fractions as separate procedures and compare the extraction used in this paper with the literature isolation procedures.
(5) Lines 131-132: How authors defined the appropriate ratios of the peptide fraction and the oligosaccharide fraction; they were 84.6% and 15.4%, respectively?
(6) Lines 136-139: Please, write the pressure that was used during the dead-off ultrafiltration process. Did the ultrafiltration fractions be characterized by the content of oligosaccharides; i.e., the DNS method, the total carbohydrate method, or the HPLC profile of oligosaccharides? what was the yield of the obtained oligosaccharides?
(7) What was the yield of the obtained peptides after enzymatic hydrolysis by use commercial food-grade endopeptidase, Alcalase 2.4 L? How you determined the protein concentration of the samples? Please, include the missing data and responses in your Materials and Method section.
(8) The appropriate letters, which denote the statistical analysis of the mean values, should be included in the findings shown in the tables as well as on the diagrams. According to the comparative approach you used, support these findings with statistical information.
(9) Figure 4: The results shown in diagrams 4a and 4b are not correct. In order to speak about enzymatic kinetics, and the course of enzymatic hydrolysis (which implies the action of an enzyme on a certain substrate), it is necessary that the value on the ordinate be zero at the initial moment (t = 0 min). Rather than subtracting whatever is present in the initial untreated sample at each point of hydrolysis, the contribution of enzyme action is measured.
(10) It is advised that the authors recheck the main text during the revision to make this manuscript more readable.
